# Ultrasound-Based Technologies for the Evaluation of Testicles in the Dog: Keystones and Breakthroughs

**DOI:** 10.3390/vetsci10120683

**Published:** 2023-12-01

**Authors:** Claudia Bracco, Alessia Gloria, Alberto Contri

**Affiliations:** Department of Veterinary Medicine, University of Teramo, Località Piano d’Accio, 64100 Teramo, Italy; cbracco@unite.it (C.B.); acontri@unite.it (A.C.)

**Keywords:** dog, testis, ultrasonography, B-flow, doppler, CEUS, sonoelastography

## Abstract

**Simple Summary:**

In andrology, ultrasound provides important information on the patient’s state of health. In dogs, as in humans and other mammals, many reproductive pathologies can affect the male’s general and reproductive health. In today’s society, the dog represents a full member of the family unit, thus early and timely diagnosis represents an important step in the treatment and resolution of the testicular pathological process, improving both reproductive and general health. Ultrasound represents a noninvasive diagnostic technique and is well tolerated by patients. Several new ultrasound-based technologies developed in recent years have expanded the tools available to the andrologist. The present review aims to describe all ultrasound techniques applied to canine testis evaluation, including the well-known basic B-Mode ultrasonography and colour Doppler, and new and advanced technologies, such as contrast-enhanced ultrasonography (CEUS) and ultrasound elastography. The principles of the different technologies, their applications, and the relevant findings in normal and abnormal testicular conditions, often completed by images, are described and discussed. Thus, the present review, describing the ultrasound-based tools available to canine andrologists, promotes the diffusion of advanced technologies for the rapid identification of canine testicular disease, promoting the chances of a resolution and restoration of reproductive function.

**Abstract:**

Ultrasonography is a valuable diagnostic tool extensively used in the andrology of human and domestic animals, including dogs. This review aims to provide an overview of various technologies based on ultrasound, from the basic B-Mode ultrasonography to the more recent advancements, such as contrast-enhanced ultrasonography (CEUS) and ultrasound elastography (UEl), all of which are utilized in the evaluation of canine testicles. The review outlines the principles behind each of these technologies and discusses their application in assessing normal and abnormal testicular conditions. B-mode canine testicular ultrasonography primarily focuses on detecting focal lesions but has limitations in terms of objectivity. Other technologies, including Doppler ultrasonography, B-Flow, and CEUS, allow for the characterization of vascular patterns, which could be further measured using specific applications like spectral Doppler or quantitative CEUS. Additionally, ultrasound elastography enables the assessment of parenchyma stiffness both qualitatively and quantitatively. These ultrasound-based technologies play a crucial role in andrology by providing valuable information for evaluating testicular function and integrity, aiding in the identification of pathological conditions that may impact the health and quality of life of male dogs.

## 1. Introduction

An essential part of the andrological examination of dogs involves the evaluation of the reproductive tract using ultrasonography [1]. Evaluating canine testes holds significant importance for both veterinary clinics and breeders selecting stud dogs; combining the strengths of traditional and modern techniques can yield a more accurate and thorough assessment of a dog’s testicular health. This approach ensures that veterinarians can provide comprehensive care and that breeders or owners can make informed decisions to properly manage their dog.

The imaging modality of choice for the evaluation of the reproductive system in both humans and animals is ultrasonography [2], which is also used for the detection and characterization of testicular lesions [3]. Ultrasonography is safe and minimally invasive, easy and ready to use, rapid to perform and interpret, and less expensive compared with other technologies, such as magnetic resonance or computerized tomography [4]. The technological advancement in this field has been very fast in recent years, adding to the conventional and well-known ultrasound methodologies, like B-mode ultrasonography and Doppler-based ultrasonography, several advanced techniques, such as contrast-enhanced ultrasonography (CEUS) and ultrasound elastography (UEl). As a consequence, the technological implementation has expanded the diagnostic armamentarium available for the andrologist, increasing the chances of an effective and rapid diagnosis [2].

This review of both conventional and innovative ultrasound-based techniques presents a description of the technology, the procedures for examination of the testis, and the typical findings in normal and pathologic testis, recognizing the strengths and limits of each technology. The information reported allows the andrologist to choose the technique or the combination of techniques most helpful and powerful to define the testicular disease and to undertake the therapeutic procedure to restore canine reproductive and general health.

To achieve this, two research databases, Scopus (https://www.scopus.com, accessed on 15 March 2023) and PubMed (https://pubmed.ncbi.nlm.nih.gov/, accessed on 16 March 2023) were consulted using systematic keywords (dog, testis, ultrasound) and, depending on the technology considered, specific keywords (B-mode, Doppler, B-flow, contrast-enhanced ultrasonography, elastography). After introducing these criteria, a total of 102 and 136 manuscripts were detected for Scopus and PubMed, respectively. All the manuscripts were consulted, and the main findings are reported in the present review. When studies were not available on the dog or when few references were available, relevant references regarding other mammals were considered for each technology.

## 2. Testes Component Evaluation by Ultrasonography

### Anatomy, Physiology and Vascularization of the Testis

Testes are the primary reproductive organs of male dogs [5]. They are located in the inguinal region within the *scrotum* and are nearly globous with a dorsal–caudal orientation of their major axis. The standard volume dimension of a canine testis was 3 cm (major axis) or 2 cm (minor axes) in an 11 kg dog, [6], but wide variability in canine body weight in different breeds, which results in large variability in testicular size, is present in this species. In smaller breeds, the testes may be relatively smaller and closer to the body due to the limited space within the scrotum. In contrast, in larger breeds, the testes may be larger and situated further away from the body.

The extra-abdominal position of the testes makes these organs easily accessible for ultrasonography evaluation, as well as the epididymis and the distal part of the spermatic cord [6].

The *epididymis* is adherent to the testis, with its head located at the testicular cranial pole of the testicle. The body runs dorsolaterally, and the tail is anchored at the dorsal-caudal extremity of the testis by the short, thick proper ligament. The tail of the epididymis is continuous with the ductus deferens, which forms the spermatic cord along with the vascular compartment. It moves within the inguinal canal and enters the abdomen via the vaginal ring [6].

The testis is composed of stroma, the connective tissue framework, and the parenchyma, which is composed of the seminiferous tubules. The stroma is formed by the external tunica albuginea, from which septa branch out to compartmentalize the parenchyma into lobules [6]. The septa merge centrally at the *mediastinum testis*, a cord of connective tissue running lengthwise through the middle of the testis [6].

Testicular blood vessels and lymphatics enter and exit through the mediastinum. The testicular artery supplies the testis, while the artery of the ductus deferens supplies the epididymis [6].

At the level of a transverse plane through the fourth lumbar vertebra, the testicular artery emerges from the ventral side of the aorta. The right artery begins cranially to the left, in line with the locations of the testicles during foetal development. Following the ductus deferens into the spermatic cord to the level of the epididymis, the artery of the ductus deferens, which is a branch of the prostatic artery from the internal pudendal, runs along the prostatic artery. It connects to the testicular artery and the epididymis through anastomoses [6].

Within the spermatic cord, the testicular vein surrounds the testicular artery, lymphatics, and nerves in an extensive pampiniform plexus, following the arterial pattern. The right testicular vein drains into the caudal vena cava at the point where its arterial counterpart originates. The left testicular vein carries blood to the left renal vein [6].

The testicular vessels are unusually long and tortuous, which is functional in creating a low temperature via thermic dispersion and exchange [7]. It also mantains a low oxygen-tension environment [8] due to the low intratesticular capillary pressure, both of which are beneficial for spermatogenesis [9].

## 3. Grey-Scale Ultrasonography

### 3.1. Technology and Applications

Grey-scale ultrasonography was the first technique applied to evaluate the testis, starting in the early 1990s. Several authors have confirmed its usefulness in providing clinicians with information about the health of the reproductive tract, making it an integral component of the breeding soundness evaluation (BSE) for dogs [10,11,12,13,14]. Grey-scale ultrasonography offers fine anatomical details of the testicle and surrounding structures [15], revealing lesions that may be too small or inaccessible for detection through palpation [1]. However, ultrasonographic changes are not specific enough to identify the different types of testicular lesions [3]. Moreover, a series of ultrasonographic scans can be very beneficial in assessing the progression of a disease and the effectiveness of treatment [16].

Grey-scale ultrasonography is a subjective procedure that allows for a qualitative evaluation of testicular parenchyma, enabling the detection of intraparenchymal lesions. However, it does not discriminate between different types of lesions or allow for a quantitative evaluation useful for assessing alterations involving the whole parenchyma without specific lesions. To enhance the use of grey-scale ultrasonography, some authors have proposed the objective estimation of the echotexture based on pixel-intensity analysis [17]. The image display consists of an array of picture elements (pixels), with each pixel representing a determined acoustic impedance displayed in a range of shades of grey (ranging from white to black) [17]. Acoustic impedance is, in turn, related to the tissue density crossed by the acoustic beam, resulting in a larger (shift toward white in the sonogram) or lesser (shift toward black in the sonogram) echo. Assuming a relationship between pixel intensity and a specific point’s ability to reflect the acoustic beam as a sign of tissue density, several studies have investigated the relationship between the pixel intensity of testicular sonograms, quantified by image-analysis software, and semen quality in domestic animals [17,18,19,20,21,22] and humans [23,24,25,26]. These studies revealed that changes in testicular pixel intensity were correlated with the percentage of morphologically normal live spermatozoa.

Likewise, although ultrasound imaging cannot establish a cytologic or histologic diagnosis, ultrasound-guided tissue sampling can be performed quickly, accurately, and safely [27] using techniques such as testicular biopsy or testicular fine-needle aspiration [28].

### 3.2. Examination Technique

As a result of the short distance between the probe and the testis, high-frequency and linear transducers should be used whenever possible, typically in the range of 7.5 to 10 MHz. Low-frequency transducers may not provide sufficient resolution to detect small lesions or subtle parenchymal changes [16].

The testicular examination is a straightforward procedure. Images can be obtained in a nonsedated dog in lateral recumbency or a standing position. Clipping of the scrotal hair should be avoided, as good images could be obtained by using generous amounts of ultrasound gel [27]. The testes should be scanned in transverse, longitudinal, and dorsal planes. One testis can be used as a standoff to image the opposite one [16]. Both testes can often be imaged in a single transverse or dorsal section, which is helpful for direct comparison [16].

### 3.3. Normal Findings

The canine testis appears echogenic with a homogeneous, medium echotexture. The parietal and visceral tunics form a thin, hyperechoic peripheral echo. The mediastinum testis is visible as an echogenic central linear structure on the midsagittal plane and as a central focal echo on a mid-transverse scan plane (Figure 1) [16].

In prepubertal dogs, the testes tend to have a more hypoechoic echogenicity compared to adult dogs, and the mediastinum testis can be easily identified [29]. Typically, there are no noticeable differences in echogenicity between the right and left testis [22,29].

The tail of the epididymis generally appears isoechoic when compared to the testicular parenchyma (Figure 2). Additionally, the tail exhibits a coarser echotexture than the testis. On the other hand, the head and body of the epididymis are nearly isoechoic with the testis. The head is located cranially, and the body can be traced caudally in both sagittal and transverse planes to reach the tail, which is consistently the most clearly imaged portion of the canine epididymis [16].

In older dogs, small hyperechoic foci representing the testicular septa are occasionally visible [30].

Testicular volume is positively correlated with total sperm count, sperm motility, sperm morphology, and daily sperm production in dogs [31,32]. Estimating testicular volume can be useful in demonstrating asymmetry or reduction, as some authors have reported that testicular volume is age-related, with the maximum size reached at 6 years, followed by a progressive decrease [33]. Ultrasonography is generally considered the most accurate method for quantitatively determining testicular volume [16]. Few studies have evaluated ultrasound mensuration of dog testicles, establishing its reliability in comparison with the results obtained using calipers (orchidometer). Among the most common formulas used for calculating testicular volume, Lambert’s formula (volume = *length* × *width* × *height* × 0.71) provides a more accurate estimate [31,32].

### 3.4. Abnormal Findings

#### 3.4.1. Intratesticular Diseases

Testicular *neoplasms* are the most common tumours of the male dog’s genital tract [27,34,35,36,37,38,39], with a prevalence of up to 60% and an incidence that increases with age [3,40,41,42,43] and cryptorchidism [37,38,39,40,44,45,46].

The ultrasonographic features of testicular tumours can vary widely [3], from small nodules distinguishable within the parenchyma to heterogeneous echotexture that alters the normal canine testicular pattern. Although testicular tumours cannot be discriminated based on shape, margin, and echotexture [3], small and well-defined hypoechoic focal lesions are usually associated with interstitial cell tumours, while large lesions, sometimes enlarging the testis, with heterogeneous echotexture are typically detected in Sertoli cell tumours or seminomas [27].

Areas of haemorrhage and necrosis can occur in all types of tumours and may be observed ultrasonographically as disorganized hyperechoic and hypoechoic regions. Additionally, areas of calcification within the testicular parenchyma can be visible, appearing as hyperechoic foci producing acoustic shadowing [27].

While testicular tumours are rarely malignant, with a metastasis rate of lower than 15%, among the testicular tumours Sertoli cell tumours are most prone to metastasize [43]. However, seminomas and Leydig cell tumours can also develop metastasis [47,48,49]. Metastasis typically occurs first in the iliac, para-aortic, and sublumbar lymph nodes. Metastasis to the liver, lungs, kidneys, spleen, adrenals, pancreas, skin, brain, and eyes has also been reported [50]. Early detection through ultrasonography allows for the orchiectomy of the affected testis, improving the chance of maintaining the patient’s fertility, especially in breeding dogs [3].

There are few reports in which dogs have been diagnosed with neoplasms of testicular origin in an extratesticular location [51]. Few possibilities, including the presence of embryological ectopic tissue or the presence of testicular tissue transplanted during castration, are considered causal factors [51]. The location of the extratesticular testicular tumours in dogs varies and includes the spermatic cord, the inside of the scrotal skin, or the site of the prescrotal castration incision site [51]. Most of the neoplasms are small, typically about 1.5 cm in diameter [35]. Apart from their location, their appearance is indistinguishable from their intratesticular counterpart [35].

*Orchitis,* as inflammation of the testis, can occur acutely or chronically. Acute orchitis may present with variable ultrasonographic characteristics, ranging from irregular and poorly defined anechoic areas to a diffuse patchy hypoechoic echo pattern, and focal abscessation may be evident [1,16,29]. Typically, the testis and epididymis enlarge, and fluid may accumulate between the visceral and parietal tunic within the scrotum [1,16,29,52]. Chronic orchitis is less obvious in terms of ultrasonographic features and may reveal hyperechoic or mixed echogenic parenchyma, often associated with a reduction in testicular size [16]. In the case of chronic progression of orchitis, abscess formation may occur, characterized by an irregular hyperechoic wall and anechoic to hypoechoic central contents [16].

*Testicular hypoplasia* is a developmental defect of the testis, preventing it from reaching the normal postpubertal size. Most cases of hypoplasia are due to cryptorchidism and are often linked to the underdevelopment of the epididymis as well [35].

In contrast, *testicular atrophy* is used to describe normally developed testes that have become smaller in size due to ageing [35], cryptorchidism, testicular tumour, or chronic orchitis in the opposite testicle [16]. An atrophic testis typically retains a normal-sized epididymis, so proportions change with increased severity.

In both testicular hypoplasia and atrophy, ultrasonography reveals thickened albuginea, with less obvious or missing blood vessels. The echotexture can vary, ranging from hypoechoic to isoechoic, or it can be diffusely hyperechoic, depending on the cause and severity [16]. Hyperechoic foci causing acoustic shadowing may also be present, reflecting parenchymal mineralization [35].

Ultrasonography is a sensitive diagnostic tool for *cryptorchidism*, used to locate and evaluate undescended testicle(s), which may be located in the abdominal cavity (sensitivity of 97.7%), inguinal canal (sensitivity of 100%), or in an ectopic subcutaneous location between the superficial inguinal ring and the scrotum [53]. This method facilitates the location of retained testes before surgical exploration or laparoscopy [53]. Cryptorchid testes are typically smaller in size, and the testicular parenchyma can often be detected by the presence of the hyperechoic mediastinum testis. Retained testicles may undergo neoplastic transformation [54], and they are usually located based on their increased size and abnormal architecture (Figure 3). They appear as masses with mixed echogenicity and varying diameters, often situated in abdominal region [14].

Occasionally, the ultrasonographic examination may reveal the presence of hyperechoic nonshadowing spots within the testicular parenchyma. These lesions have been referred to as *testicular microlithiasis*, a condition well defined and described in human andrology [55]. Unfortunately, in domestic animals, this condition has been poorly characterized, and the epidemiology, ultrasonographic appearance, as well as its biological significance and implications, remain largely understudied in both large and small animals.

#### 3.4.2. Extratesticular Diseases

*Epididymitis* can occur separately or concurrently with orchitis (Figure 4), and the damage may extend to include the ductus deferens [1,16,29]. Typically, epididymitis involves the tail and sometimes the body of the epididymis, with the head of the epididymis seldom being affected [56].

This condition may be bilateral or unilateral, with varying severity that reflects the degree of damage, including necrosis and vascular changes. In cases of severe acute disease, there is swelling and oedema of the tail of the epididymis, resulting in a relevant increase in size [35], sometimes accompanied by fluid accumulation into the vaginal cavity. Ultrasonography, through direct visualization of the altered organs, aids in the differential diagnosis of diseases that cause scrotal volume increase [14,16,29,56].

*Torsion of the spermatic cord* is uncommon in dogs [16]. Due to the peculiar structure of testis vascularization, testicular necrosis can result from varying degrees of torsion, and it is more frequently observed in retained (or cryptorchid) testes [35]. Additionally, torsion of the spermatic cord in an intra-abdominal testicle has been frequently reported in the presence of testicular tumours [1,16,44]. The ultrasonographic appearance of experimentally induced testicular torsion in the dog has been reported by Hricak et al.: between 15 and 60 min after torsion, there are anatomical changes detectable by ultrasound, such as testicular enlargement characterized by diffusely decreased parenchymal echogenicity, concurrent enlargement of the epididymis and spermatic cord, and hypoechoic thickening of the scrotal skin [57].

Ultrasonography can be particularly valuable in cases of extratesticular (around the testicle but within the vaginal tunic) *fluid accumulation*. In this case, the fluid causes a scrotal enlargement, preventing clinical discrimination between the structures involved. Types of extratesticular fluid accumulation include serum (hydrocele), blood (haematocele), pus (pyocele), or possibly urine [12,14].

*Epidydimal cysts*, resulting from epididymal canal occlusion, are rarely reported in dogs [12].

*Spermatic granulomas* are characterized by the accumulation of spermatozoa in efferent ducts, the epididymis, or the deferent duct, surrounded by macrophages and other inflammatory cells [58]. Clinically (hard nodular lesion) and on grey-scale ultrasonography, this lesion can be challenging to differentiate from neoplasia. Cytology via fine-needle aspiration or histology from biopsy or orchiectomy is necessary to confirm the non-neoplastic nature of the lesion [59]. Unfortunately, no reports describing spermatic granulomas in dogs are available in the literature.

*Varicocele* in humans results from alterations in the veins of the pampiniform plexus, leading to enlargement, elongation, and tortuosity. On sonograms, varicocele appears as anechoic, tubular, and serpiginous fluid collection in the region of the epididymis. Varicocele is rarely encountered in dogs [12,60]. It is characterized by the formation of varicose veins in the scrotal region. Ultrasonographic imaging reveals vascular dilatation [61].

## 4. Colour Doppler and Power Doppler

### 4.1. Technology and Applications

Doppler ultrasonography has become the method of choice for evaluating the blood supply of the testis. It is one of the simplest and most precise techniques for estimating blood flow, as it combines data concerning the anatomy and dynamic flow parameters [62].

Based on how blood flow information is displayed, Doppler ultrasonography can be classified as colour (colour Doppler and power colour Doppler) or spectral (pulsed wave—PW, continuous wave—CW), or a combination of both.

Colour and power colour Doppler are termed colour Doppler due to the use of colour map overlays on real-time two-dimensional grey-scale images to visualize blood flow (Figure 5) [16]. These overlays represent signals from moving red blood cells in colour, indicating the direction of their motion toward or away from the transducer. The amount of colour saturation also conveys information about the relative velocity of cells.

Power colour Doppler ultrasonography is more sensitive in detecting low velocities and small parenchymal vessels [16,63]. Due to the qualitative nature of the interpretation of sonograms, colour Doppler is rarely applied in experimental studies.

Quantitative blood flow analysis includes the evaluation of peak systolic velocity = PSV, end-diastolic velocity = EDV, resistance index = RI, and pulsatility index = PI.

Pulsed-wave and continuous-wave Doppler are collectively referred to as spectral Doppler. They display quantitative information in the form of time-velocity waveforms along the *y* and *x* axes, respectively. Pulsed-wave Doppler ultrasonography transmits sound in pulses using the pulse-echo principle, similar to real-time imaging [16]. Pulsed-wave Doppler is the most commonly used type of spectral Doppler because it is readily available on nearly all modern transducers and provides depth discrimination [16]. Continuous-wave Doppler technology can measure much higher flow velocities than pulsed Doppler [16,64].

Duplex Doppler ultrasonography involves the simultaneous display of pulsed- or continuous-wave spectral Doppler tracings and B-mode images [16].

Similarly, triplex Doppler ultrasonography combines two-dimensional ultrasound, colour Doppler, and pulsed Doppler, allowing for the collection of anatomical data of the vessels and functional data regarding blood flow, including its presence or absence, direction and speed [16,65].

In human medicine, pulsed-wave Doppler, colour, and power Doppler are routinely applied to asses andrology status [56,66,67,68,69,70,71,72,73,74,75,76] and to determine the aetiology of dyspermia [23,77], demonstrating that sperm quality and quantity are dependent on tissue perfusion [23,26,78,79,80] and suggesting that the evaluation of testicular blood flow can predict the testicular function and, in turn, spermatogenesis [23,78,79,80].

In veterinary medicine, the study of testicular vascularization has been conducted in various animal species, including stallions [8,81,82,83,84], jackasses [85], tomcats [86,87], bulls [17,88,89], rams [20,90,91], bucks [62,92], and dogs [52,65,93,94,95,96,97,98,99,100,101,102,103]. The results of these studies essentially confirm a relationship between testicular arterial blood flow and seminal quality in both normal and pathological conditions.

### 4.2. Normal Findings

The peculiar vascularization of the testis allows for the division of vessels into three segments: (a) supratesticular arteries within the pampiniform plexus, which include the testicular artery, cremasteric artery, and deferential artery; (b) arteries within the testicular membranes, represented by the marginal artery; and (c) intratesticular vessels, comprising the centripetal branches and recurrent rami [99].

Characteristics of blood flow within the testicular artery, assessed by pulsed-wave Doppler ultrasound, vary depending on the segment [97,99]. In the supratesticular region, the blood flow exhibits a biphasic waveform with a diastolic notch followed by a diastolic peak, or it may display a monophasic waveform characterized by a slow systolic increase followed by decreased diastolic flow, attributed to the vessel’s tortuous nature in this area. The other two regions typically demonstrate low-resistance flow with monophasic waveforms (Figure 6) [29]. Blood flow velocities are higher in the supratesticular region, gradually decreasing through the marginal and intratesticular regions [29,94,97,99].

### 4.3. Relationship between Spectral Doppler Measurement and Dog’s Semen Quality

As mentioned earlier, there is an observed relationship between pulsed-wave Doppler measurements and semen quality in various studies, primarily in human medicine, which report strong correlations between the values of the resistive index (RI) and peak systolic velocity (PSV) with sperm production rate scores. These parameters are considered reliable indicators of spermatogenesis and are proposed for routine clinical protocols in distinguishing different causes of dyspermia and identifying subfertile men [104].

In veterinary medicine, the measurement of these indexes could be an effective tool in andrology as potential markers of seminal quality in dogs. For example, in the study of Zelli et al., these indexes were studied in correlation with testicular volume and semen parameters. The results showed a positive correlation between peak systolic velocity and testicular volume and a negative correlation with live sperm. Additionally, a negative correlation was observed between the resistive index and pulsatility index with total and progressive motility. The resistive index and pulsatility index also showed negative correlations with the percentage of membrane-intact sperms with curled tails, while the latter exhibited a positive correlation with end-diastolic velocity [103].

Moreover, in Gloria et al.’s study, the blood flow parameters measured by pulsed-wave Doppler were evaluated in correlation not only with sperm attributes but also with testicular histological characteristics. They confirmed negative correlations between RI and PI with abnormal spermatogenesis and histological abnormalities [98].

Additionally, Velasco and Ruiz proposed the use of ultrasonographic measurement as objective parameters to evaluate testicular function. However, they noted variability in all analysed data, depending on factors such as the measurement location, season, species, breed, and laterality. They concluded that further research is needed to establish physiological parameters for pulsed-wave Doppler measurements [105].

### 4.4. Abnormal Findings

In the context of testicular *neoplasia*, colour (Figure 7) and power Doppler (Figure 8) ultrasonography are valuable tools for assessing tumour vasculature. This is due to the typically high interstitial pressure in tumours and the resulting low-velocity states in tumour vessels. However, there are limited descriptions of testicular blood flow in abnormal testes [106].

The vascularity index (VI) appears to increase in solid tumours compared to non-neoplastic masses, as blood flow within and around most tumours is enhanced [52]. Blood flow PSV increases with the size of neoplastic nodules, and the spectral waveform around the lesion typically exhibits a low-resistance pattern with lower to middle values of PI and RI compared to normal tissue [52]. However, it is important to note that none of these appearances are specific to a particular tumour type [3,27,52].

In cases of *orchitis,* testicular blood flow, as estimated by colour Doppler, may demonstrate increased perfusion within the testicular parenchyma, accompanied by an increase in RI and PI [29]. However, only limited modifications in vascular flow may be detected in cases of necrosis and fibrosis [52]. The report also suggests that in inflammatory lesions, RI may significantly decrease due to reactive hyperaemia, while it increases in fibrotic lesions associated with degenerative changes.

In dogs with *torsion of the spermatic cord*, colour Doppler ultrasound can help in identifying the absence of perfusion to and within the twisted testis [29,57], aiding in the differential diagnosis of acute orchitis [12]. In cases of incomplete torsion, it may be possible to observe decreased perfusion rather than complete absence [29].

## 5. B-Flow

B-Flow is a type of digitally encoded ultrasound technology developed specifically by GE Healthcare (Chicago, IL, USA) for visualizing blood flow [107]. This technology is based on a combination of coded excitation and tissue equalization [107], allowing for the direct visualization of moving blood echoes using a grey-scale presentation that demonstrates real-time blood movement, similar to a conventional angiogram (Figure 9), along with simultaneous visualization of the surrounding anatomy [108].

In human medicine, B-flow was first applied for carotid artery ultrasound in vascular medicine. More recently it has been found valuable for assessing abdominal structures, such as hepatic vasculature and renal perfusion [108,109,110,111]. Furthermore, B-Flow imaging has been explored in obstetrics and gynaecology, focusing on utilizing the technology in perinatology and foetal echocardiography [107].

Applications of B-flow in veterinary medicine have been relatively limited. There are only two studies that have used this technology in animals, both evaluating chemically induced mammary tumours. These studies have reported that B-Flow is more sensitive than power colour Doppler in detecting tumour vessels [112,113].

In the field of veterinary clinical reproduction, B-flow technology has been primarily focused on evaluating vascular patterns in testicular neoplasm, with findings suggesting that these patterns do not significantly vary among different tumour types [3].

## 6. Contrast-Enhanced Ultrasonography (CEUS)

### 6.1. Technology and Applications

Contrast-enhanced ultrasonography (CEUS) was introduced in veterinary medicine relatively recently due to its ability to quantify microvascular blood volume and flow within vital organs, similar to its application in human medicine [114,115]. CEUS is based on the intravascular injection of specific ultrasound contrast agents (USCAs), consisting of microspheres containing gases stabilized by an outer shell [116,117]. Gases are eliminated through the lungs, while the stabilizing components are filtered by the kidneys and eliminated by the liver [86,118,119]. These microspheres reflect ultrasound echoes, significantly increasing the intensity of the signal in grey-scale and Doppler modes, enhancing visualization for approximately 5 min, depending on the contrast agent used. A complete CEUS evaluation typically takes around 20 min and does not require general anaesthesia [119].

The growing adoption of CEUS in diagnostics is attributed to its safety, painlessness, speed, portability, and lack of irradiation. It does not have nephrotoxic effects, and side effects from USCAs in dogs are rare, with reported cases limited to injection-site pain, nausea, or vomiting [120]. In spite of the costs of the contrast material and the need for specialized ultrasonographic equipment [29], CEUS is considered relatively affordable when compared to computed tomography and magnetic resonance imaging [117,121] and can be performed on awake patients without requiring general anaesthesia [119].

CEUS enables the detection of several lesion attributes, including the presence or absence of contrast, wash-in (incoming phases) and wash-out (output), peak enhancement, temporal behaviour, perfusion characteristics, vascular anatomy, comparison with the surrounding tissues, and flow direction [86,117,122]. Some of these attributes can be quantitatively estimated within a specific region of interest (ROI) using the integrated software in ultrasound machines [3], which is able to calculate peak intensity (PI), time to peak (TTP), and area under the curve (AUC). The wash-out (WO), as the time interval from TTP until signal intensity declined by 40% of PI, could be also calculated [3].

Over the past decades, both ultrasound contrast agents and techniques have evolved rapidly. CEUS is now used to quantify perfusion in various deep organs, including skeletal muscle, the heart, adipose tissue, kidneys, liver, and brain [114]. Recent studies have applied CEUS in human testis evaluation, investigating various features in different pathological conditions [4,123,124,125,126].

In veterinary medicine, CEUS has primarily focused on dogs, especially in assessing the liver and its vascularization [116,127,128,129,130,131,132], lymph nodes [129,133,134], kidneys [135,136], pancreas [137,138], eyes [139,140,141], spleen [117,142,143,144,145], and prostate [146,147,148,149,150]. These studies have examined normal and pathological aspects, differentiating between inflammatory, degenerative, and neoplastic lesions, while providing discrimination between benign or malignant conditions [106,119,151]. Some studies have also been performed on cats [86,152,153]. A limited number of studies have explored the application of CEUS in canine andrology, primarily focused on chronic testicular alterations [120].

### 6.2. Normal Findings

The contrast agent, after injection, enables the observation of the branching of the testicular artery and parenchymal perfusion, with a progressive opacification of the convoluted supratesticular, marginal, and then intratesticular arteries, with flow directed towards the mediastinum testis. A progressive increase in the echogenicity of the testicular parenchyma during the vascular bed phase is observed, followed by gradual clearance of the contrast from the parenchyma during the wash-out phase. Testicular veins are also highlighted, with lower echogenicity compared to the arteries, as they exhibited a longer wash-out period due to their persistence within the vascular bed [120]. It is important to note that a range of values for CEUS in normal dogs was proposed. However, reference values should be used with caution due to the lack of reproductive information from the healthy group and the sedation of the patients during the procedure, which could alter the physiological vascular flow.

### 6.3. Abnormal Findings

Due to the detection of fine vascularization, CEUS can detect some testicular lesions not previously revealed by conventional ultrasonography, resulting in a particularly powerful assessment of testicular neoplasia [3,120]. *Neoplastic lesions* have been described to be better defined in the wash-in phase and tend to maintain the pattern during peak and wash-out phases. Thus, specific phases are not associated with different patterns of contrast enhancement in lesions over time [120]. According to a previous study, CEUS was a highly efficient technique for detecting both neoplastic and non-neoplastic testicular lesions, with a sensitivity of 87%, specificity of 100%, a positive predictive value of 87%, and a negative predictive value of 100% [117].

Although some authors have hypothesized an association between specific CEUS patterns and different tumour types, the results have been somewhat controversial [3]. Volta et al. reported the association between hypo- or iso-enhanced testicular lesions with intralesional vessels with seminomas [120], while Orlandi et al. found similar CEUS parameters in different tumour types, suggesting the inability to differentiate between testicular tumours based on their contrast-enhanced pattern [3].

A CEUS pattern in *non-neoplastic testicular lesions* was reported in a study on dogs, which included degenerated testes, atrophic testes, testes with chronic necrotic orchitis, and testes with interstitial cell hyperplasia [120]. However, the limited number of cases and the lack of specific lesions in different conditions have reduced the application potential of this approach.

## 7. Ultrasound Elastography

### 7.1. Technology and Applications

Ultrasound elastography (UEl) is an ultrasonography-based technology introduced in the 1990s, designed to assess the elasticity or the stiffness of the tissues [154]. This technique is grounded in the concept that softer tissue deforms more readily under compression compared to harder tissue, enabling the estimation of tissue elasticity [155]. Tissue elasticity changes can be associated with pathological modifications induced by degeneration (ageing), inflammation, and uncontrolled cell growth [154].

Elastography utilizes ultrasonic imaging to monitor tissue shear deformation under conditions of one or both shear conditions, often in real-time two-dimensional image sequence, following the application of dynamic forces (e.g., thumping or vibrating) or gradual forces considered “quasi-static” (e.g., probe palpation). The deformation can be represented in an elastogramme, or as a local measurement, in one of three ways: (i) tissue displacement may be directly detected and displayed, as in the method known as acoustic radiation force impulse (ARFI) imaging; (ii) tissue strain can be calculated and displayed, producing what is termed strain elastography (SE); (iii) in the dynamic case only, data can be used to record the propagation of shear waves, which can be employed to calculate regional values of their speed (without creating images) using methods referred to herein as transient elastography (TE) and point shear-wave elastography (pSWE), or images of their speed using methods referred to herein as shear-wave elastography (SWE), which includes 2-D SWE and 3-D SWE [156,157].

Various technologies can evaluate tissue stiffness qualitatively, semi-quantitatively, or quantitatively. Strain elastography, for instance, translates the deformation of tissue resulting from manual probe compression into a grey-scale image. The final result is an image, namely an elastogramme, in which the stiffness of the different components of the anatomical region is displayed in real time on a colour-coded map [158]. Due to the subjective origin of the compression, this technique could be considered qualitative. To increase objectivity, a semi-quantitative measurement has been proposed, involving the calculation of the strain ratio. This ratio compares the stiffness of the tissue under examination, as determined in a region of interest (ROI), to that of adjacent normal tissue in a similar-sized ROI [157].

In contrast to strain imaging, shear-wave elastography allows the measurement of the speed of shear-waves generated by the probe, enabling the quantitative assessment of tissue elasticity even within a specific range of interest (ROI) [157].

In human medicine, UEl has found success in diagnostic imaging of various organs, such as the liver, breast, prostate, thyroid, and lymph nodes and musculoskeletal pathological conditions. It is widely used in oncology for predicting lesion malignancy, including in cases of testicular neoplasm [155,158,159,160,161,162]. In human andrology, UEl has also been proposed for the examination of nonfocal alterations of the testis, aiding in infertility aetiology determination [26,163,164].

In veterinary medicine, studies have been conducted to apply UEl to canine and feline normal and pathological conditions in various organs, including the spleen, liver, kidney, prostate [165,166,167,168], adrenal glands [169], skin [170], lymph nodes [133,171], pancreas [172], small intestine mucosa [173], and placenta [174]. The qualitative study on foetal lungs and liver during the final days of intrauterine development were also reported, revealing tissue stiffness changes [175]. In veterinary oncology, UEl has been considered a complementary diagnostic tool to distinguish benign from malignant lesions, mainly focusing on mammary neoplasm, with inconsistent results. Some preliminary studies reported that the cyto-/histologic mammary lesions classified as benign are observed as deformable, whilst malignant exhibited rigidity, resembling the characteristics of breast tumours of women [176,177]. Contrariwise, a more recent work reported that malignant mammary nodular lesions showed stiffness similar to hyperplastic/benign neoplastic lesions [178]. Therefore, further investigations into tissue mechanical properties are needed to optimally incorporate the use of this technology in the evaluation of mammary gland tumors in dogs.

Limited information is available about the use of UEl in veterinary andrology [167,177,179,180], especially in cases of no focal alterations.

### 7.2. Normal Findings

In normal canine testes, qualitative (Figure 10) and quantitative ultrasound elastography revealed firm, uniform and not pliable organs.

Consistent with the findings in tomcats [179], shear velocity values were similar in animals grouped by age, suggesting that age had a limited effect on testicular stiffness (Figure 11) [167].

The reference values presented in this manuscript, however, should be interpreted with caution due to the variability in the breeds and weights of the males recruited and the absence of information about their reproductive function.

### 7.3. Abnormal Findings

Limited information is currently available regarding the use of UEl in the assessment of testicular disease in dogs. In a recent study, UEl was found to have the potential to differentiate between non-neoplastic and neoplastic testicular lesions, with the latter exhibiting greater stiffness [181]. Although the findings from this study were not conclusive due to the limited number of animals, ultrasound elastography shows promise as a technology for evaluating neoplastic lesions in canine testis.

In a preliminary study on UEl in canine testicular assessment, abnormal testes were observed to be stiffer and more heterogeneous compared with normal testes, even though there was a great variability among the different testicular diseases, making it challenging to compare these conditions effectively [180].

## 8. Conclusions

This review has examined the literature on the use of ultrasound-based technologies for the assessment of the canine testis. Conventional technologies, such as B-mode ultrasonography, have shown their effectiveness in andrological practice, but the lack of objectivity limits the relevance in differentiating testicular disease. On the other hand, emerging technologies, such as CEUS and ultrasound elastography, have the potential to expand the armamentarium for the clinicians caring for canine andrological patients. Still, further studies and solid results are needed to establish their real roles and contributions to clinical practice.

## Figures and Tables

**Figure 1 vetsci-10-00683-f001:**
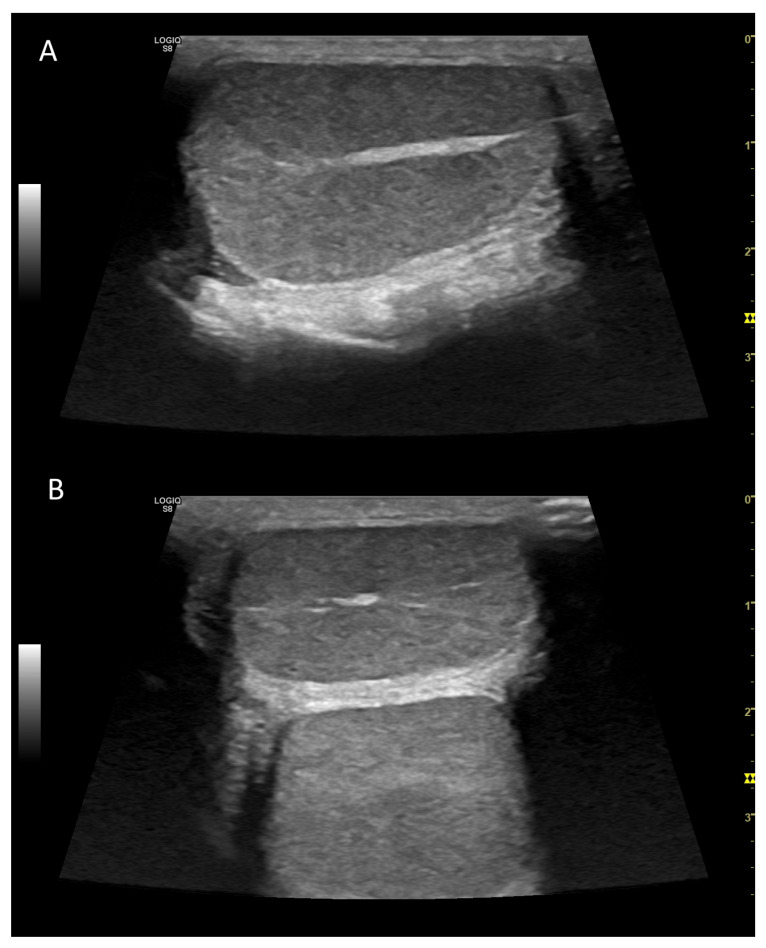
Example of B-Mode ultrasonography of normal testis in the dog, in longitudinal (**A**) and transversal (**B**) scan. The hyperechoic mediastinum is visible as a band (**A**) or a circular area (**B**) within the homogeneous parenchyma.

**Figure 2 vetsci-10-00683-f002:**
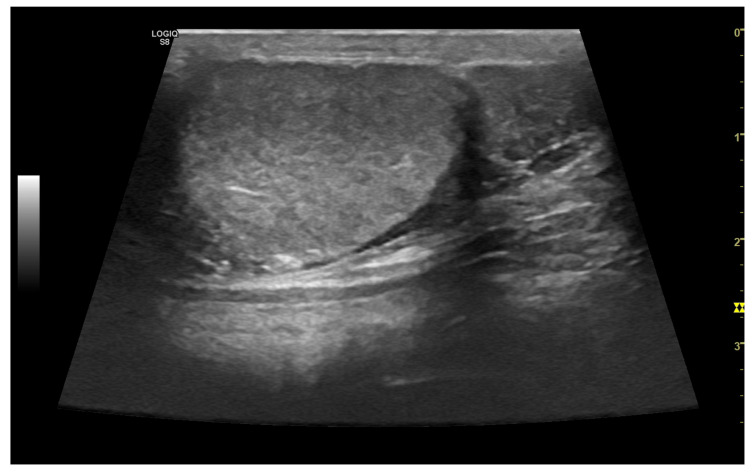
Example of B-Mode ultrasonography of normal epididymis in the dog. The normal tail of the epididymis, isoechoic, can be detected on the **right** of the image, closely adherent to the testis (on the **left**).

**Figure 3 vetsci-10-00683-f003:**
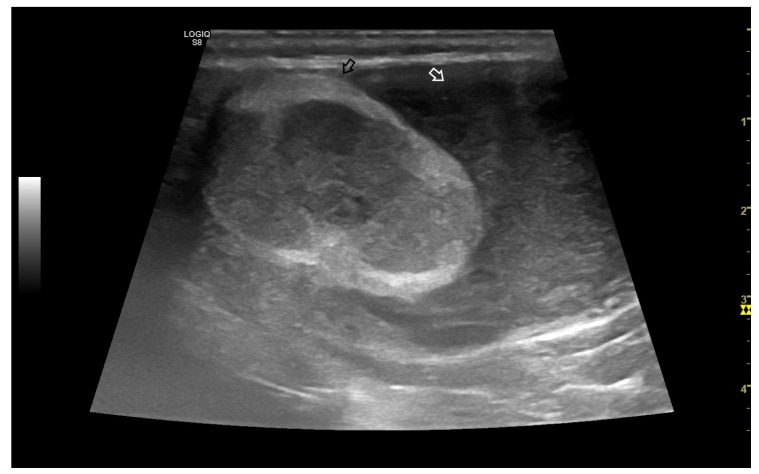
Ultrasonographic B-mode appearance of a large tumour in the testis (black arrow) compared to the testicular parenchyma (white arrow) in an inguinal cryptorchid testis. Based on histology, the image is consistent with a Sertoli cell tumour.

**Figure 4 vetsci-10-00683-f004:**
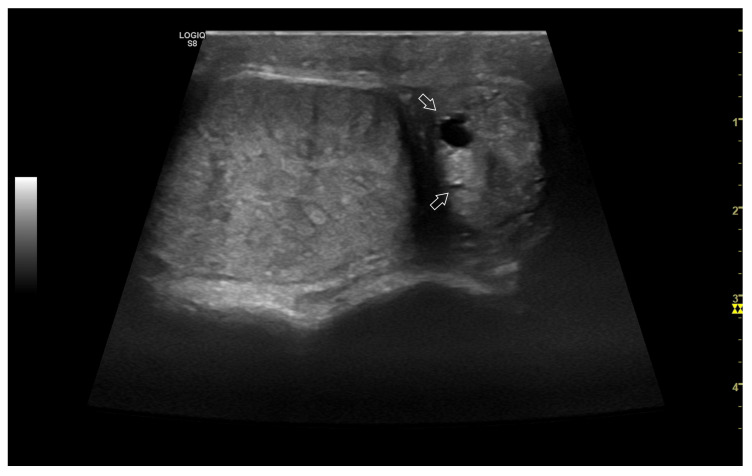
A case of epididymitis, contextual with orchitis, in the canine testis involving the tail of the epididymis. Note the relative hypertrophy of the tail of the epididymis (on the **right**) compared to the correspondent testis (on the **left**). Both organs appear heterogeneous, and small anechoic areas can be detected (white arrows).

**Figure 5 vetsci-10-00683-f005:**
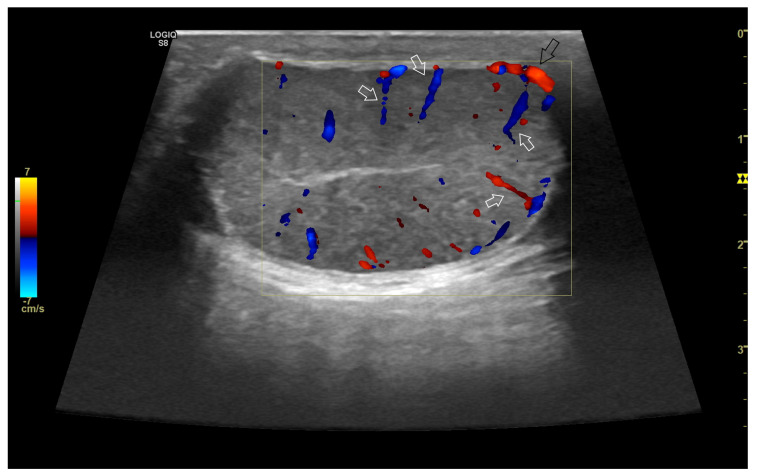
Representative sonogram in colour Doppler of canine testis. Note the marginal portion (black arrow) and the intratesticular branches (white arrows) of the testicular artery.

**Figure 6 vetsci-10-00683-f006:**
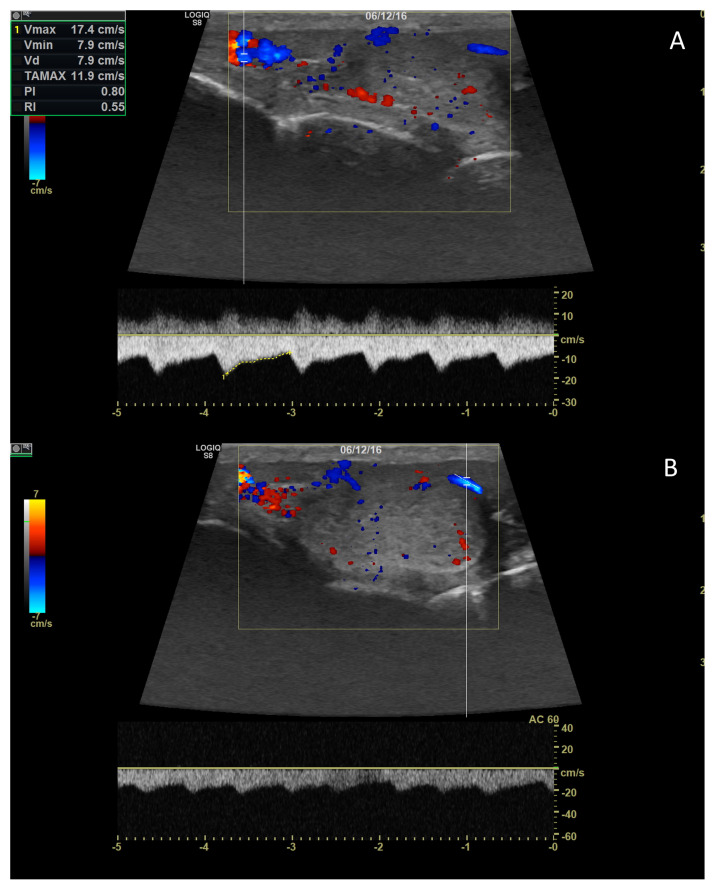
Pulsed-wave Doppler parameters were measured on the supratesticular (**A**) and marginal (**B**) portions of the testicular artery.

**Figure 7 vetsci-10-00683-f007:**
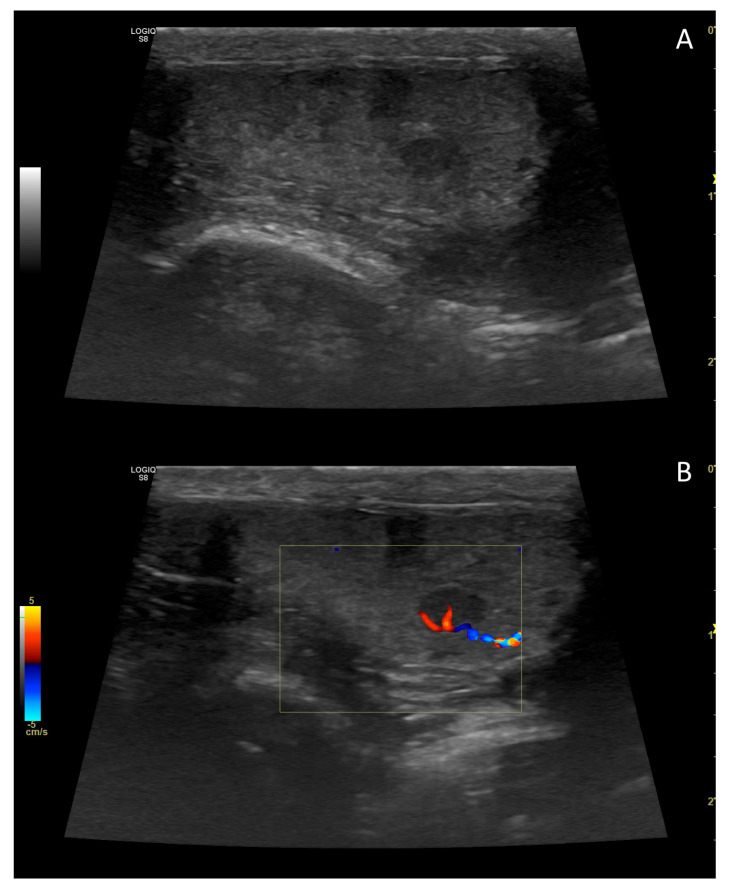
B-mode (**A**) and colour Doppler (**B**) sonogram of a small testicular tumour in the dog. Based on histology, the image referred to an interstitial cell tumour.

**Figure 8 vetsci-10-00683-f008:**
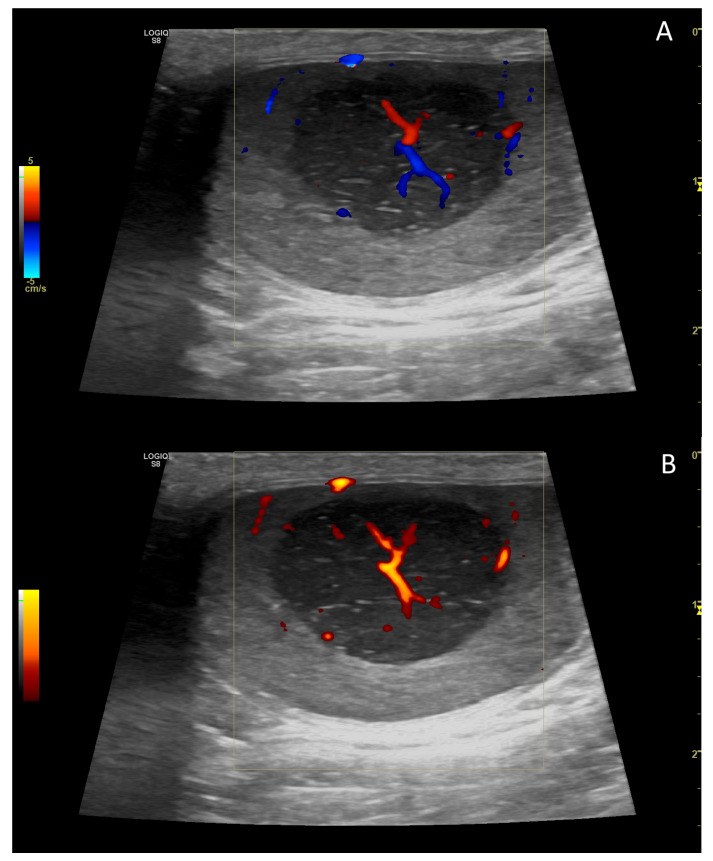
Colour (**A**) and power Doppler (**B**) sonogram of testicular tumour in the dog. Based on histology, the image referred to a mixed Sertoli cell/seminoma tumour occupying most of the testis.

**Figure 9 vetsci-10-00683-f009:**
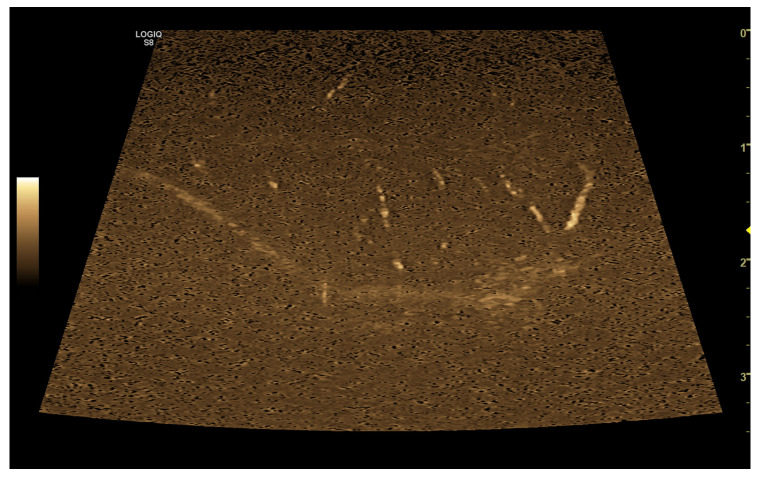
Representative image of the B-Flow evaluation of canine testis.

**Figure 10 vetsci-10-00683-f010:**
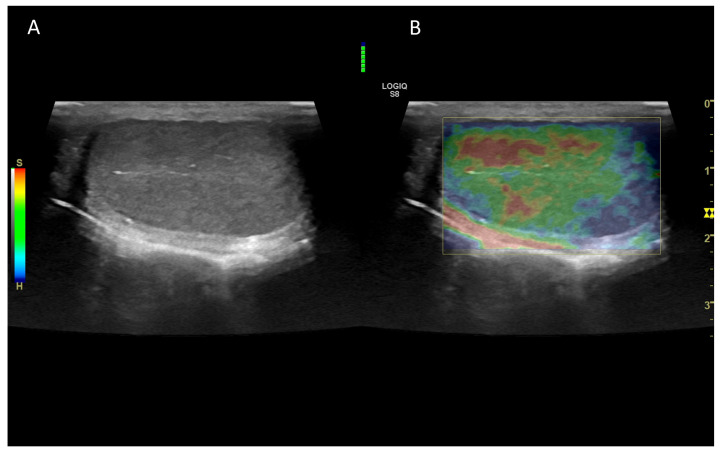
Representative elastogramme acquired with the qualitative strain elastography of canine testis (**B**) compared with B-Mode ultrasonography (**A**). The colour scale differentiates the hard (blue) and soft (red) area of the testis. The green bar in the centre of the image represents the adequateness of the freehand compression applied by the operator.

**Figure 11 vetsci-10-00683-f011:**
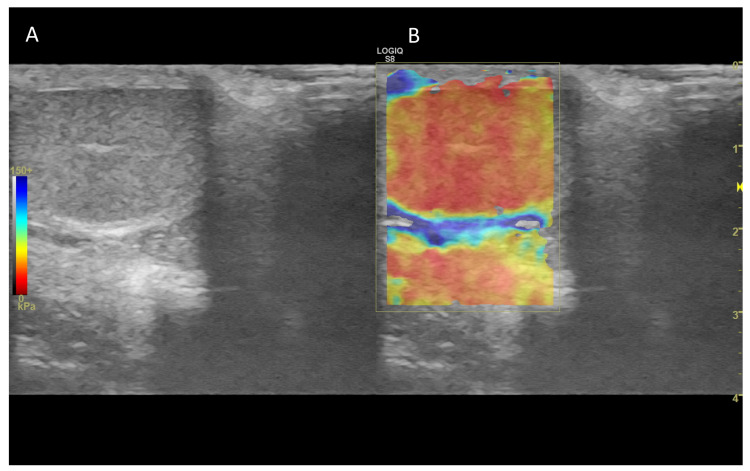
Representative elastogramme acquired with the quantitative shear-wave elastography of canine testis (**B**) compared with B-Mode ultrasonography (**A**). The technology can quantify the stiffness objectively.

## Data Availability

Not applicable.

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
