# Peer review of "Ultrasound-Based Technologies for the Evaluation of Testicles in the Dog: Keystones and Breakthroughs"

_vetsci, 2023, doi:10.3390/vetsci10120683_

Round 1

Reviewer 1 Report (Previous Reviewer 1)

Comments and Suggestions for Authors

Dear Authors

I reviewed the manuscript entitled "Ultrasound-based technologies for the evaluation of testicles in the dog: keystones and breakthroughs. The paper is improved all my concerns were addressed".

I only have a suggestion: please add a figure of a CEUS of the testes.

Therefore I recommend minor revision

Author Response

We agree with the reviewer that an image of the testicular CEUS could be an implementation for the manuscript. On the other hand, in our activity, we have not implemented this technology in our routine. In turn, we have no canine testicular (normal or pathological) images with CEUS.

Reviewer 2 Report (New Reviewer)

Comments and Suggestions for Authors

This is an interesting review manuscript related to the ultrasound-based technologies for the evaluation of testicles in the dog: keystones and breakthroughs. Despite the interest and updates, manuscript lacks a slight language review, because there are some minor language mistakes that need to be solved before acceptance. Furthermore, there are other issues that lacks to be adjusted as reported below:

1. Introduction - It is very vague. Authors should highlight the importance of evaluating canine testes for clinics or for selecting stud dogs. Them, authors are advised to highlight the importance of the present review: what new issues does it bring? How is it different from the other many reviews that we find at the literature? Why updating this subject is important for the field?

2. Testes components - This is a very general text. Authors should consider peculiarities found in dogs, related to position, morphometry, etc. Remember that we have small breeds as the Chihuahua and larger ones as the Mastiffs. So, size matters and should be highlighted. Please, improve your statements related to dog testes considering different breeds.

In general, these are the unique issues that needs to be solved. Manuscript is very well written in terms of contents, but it should be revised regarding the writing structure. Authors should avoid paragraphs too short, with only one sentence. 

Comments on the Quality of English Language

Despite the interest and updates, manuscript lacks a slight language review, because there are some minor language mistakes that need to be solved before acceptance. In general, these are the unique issues that needs to be solved. Manuscript is very well written in terms of contents, but it should be revised regarding the writing structure. Authors should avoid paragraphs too short, with only one sentence. 

Author Response

This is an interesting review manuscript related to the ultrasound-based technologies for the evaluation of testicles in the dog: keystones and breakthroughs. Despite the interest and updates, manuscript lacks a slight language review, because there are some minor language mistakes that need to be solved before acceptance. Furthermore, there are other issues that lacks to be adjusted as reported below:

Response: We deeply revised the manuscript with the help of an expert with skills in scientific English.

  1. Introduction - It is very vague. Authors should highlight the importance of evaluating canine testes for clinics or for selecting stud dogs. Them, authors are advised to highlight the importance of the present review: what new issues does it bring? How is it different from the other many reviews that we find at the literature? Why updating this subject is important for the field?

Response: we extensively modified the introduction to stress the usefulness of the review for the clinicians, who are informed about the techniques, the procedures, the normal and abnormal findings and the pros/contras of each method. This makes clinicians aware of the most appropriate technique to reach readily to a diagnosis and start, if possible, the appropriate therapeutic procedure.

  1. Testes components - This is a very general text. Authors should consider peculiarities found in dogs, related to position, morphometry, etc. Remember that we have small breeds as the Chihuahua and larger ones as the Mastiffs. So, size matters and should be highlighted. Please, improve your statements related to dog testes considering different breeds.

Response: We implemented the “Testes components” to be more adherent and specific to the dog. We also highlight the large variability in size in different breeds of this species. Finally,

In general, these are the unique issues that need to be solved. The manuscript is very well written in terms of contents, but it should be revised regarding the writing structure. Authors should avoid paragraphs too short, with only one sentence.

Response: We revised whole the manuscript with great attention to the short sentences.

This manuscript is a resubmission of an earlier submission. The following is a list of the peer review reports and author responses from that submission.

Round 1

Reviewer 1 Report

Comments and Suggestions for Authors

Dear Authors

I reviewed the manuscript entitled "Ultrasound-based technologies for the evaluation of testicles in the dog: keystones and breakthroughs" by Bracco et al. The manuscript is a review of the literature of ultrasound -based technologies of testicular disease, it is well written but too schematic, and my opinion in-depth analysis of certain topics is mandatory (see comments below), since there are several recent similar reviews in the literature. Therefore I recommend major revision.

Specific comments

Anatomy, physiology and vascularization of the testis :

please add a more thorough description of the origin of testicular arteries and the arteries of ductus deferens and where the testicular veins join the caudal vena cava. 

Normal findings

please add that in older dogs, small hyperechoic foci representing the testicular septa can be occasionally seen (Penninck and D'anjou, 2015).

Abnormal findings:

 please rephrase testicular neoplasms paragraph, since is very similar to de Souza et al (2017) review.  

Although testicular tumors are seldom malignant, please add a description of which tumor type more likely metastasize and where (i.e. regional lymph nodes etc)

Please add a brief description of spermatic granulomas, which may be a differential diagnosis of testicular nodules.  

Varicoceles has been described in dogs as well. Please add the reference ( de Magalhães et al Testicular ultrasound evaluation in small animal practice, 2019)

It would add more figures of focal and diffuse testicular lesions in B-mode and with Doppler and/or CEUS. Iconographic review may also be interesting to the readers. 

Reviewer 2 Report

Comments and Suggestions for Authors

The fundamental rationale of writing a review article is to make a readable synthesis of the best literature sources on an important research topic such as ultrasound of the canine testes. Use of proper methodologies in review articles is important in that readers assume an objective attitude towards updated information.

It would be advisable to state the process for selecting studies (that is, for screening, for determining eligibility, for inclusion in the systematic review, and, if applicable, for inclusion in the meta-analysis).

Write a structured summary including, as applicable, background; objectives; data sources; study eligibility summary criteria, participants, treatments, study appraisal and synthesis methods; results; limitations; conclusions and implications of key findings.

Describe methods used for assessing risk of bias in individual studies (including specification of whether this was done at the study or outcome level, or both), and how this information is to be used in any data synthesis.

In its current state, I do not recommend accepting this paper.

Comments on the Quality of English Language

Unfortunately, the language and sentence structures of this manuscript are at times not entirely comprehensible. The paper needs rewriting and thorough language editing to allow for a proper peer review.

Reviewer 3 Report

Comments and Suggestions for Authors

This paper is proposed as a review on Ultrasound-based technologies for the evaluation od testicles in the dog. To my knowledge the review topic is completely covered considering ultrasound based modalities and their description, however the paper do not discuss into detail the clinical usefulness of ultrasound based techniques in dogs (Se, Sp, PPV, NPV)

This point should be considered. 

This paper is similar to other in literature, however this is not a reliable motivation to not consider it for publication. It is clear and well-structured.

Authors propose a wide description of the topic and they propose a comprehensive analysis of the literature with a lot of references to human medicine and veterinary medicine (other species then dog).

Specific comments:

Line 25 - Ultrasonography is considered "the gold standard”, change with “the imaging modality of choice”(see the citation). Moreover this sentence refers to humans and it should be declared. Is it the same also in dogs? (Consider citation)

Line 32 - The citation refer to human medicine. Can you provide a reference for canine medicine?

Line 74-76. This sentence needs more explanations. B-mode ultrasonography is a qualitative evaluation and it is operator dependent but these are not considered as major limits for the correct interpretation of lesions. Ultrasound can detect the presence or absence of alterations present into a parenchyma rather than refer them to a specific disease (histopathology is the gold standard)

Line 78 -  Maybe the authors mean “acoustic impedance” and not “tissue density”. Please correct

Line 92 - Consider to explain the physical reason of that. This could be useful for the reader

Line 98 - Today the use of a standoff pad is obsolete. New generation ultrasound machines and probes provide excellent images without the need for a stand off pad

Line 146 - Not correct. A large complex mass should not be considered as a “diffuse tumor”. With the term “diffuse tumors” authors should refer to tumors that completely substitute the normale parenchyma or infiltrate it.

Line 155-157 - The citation for this  paper is not correct. The paper cited conclude that “ a multi parametric approach would be useful in those cases where a unilateral castration could be suggested or……..may suggest skipping surgery”. Moreover this sentence sounds like a personal opinion of the authors that is not widely approved/shared by the  oncologic literature in dogs. Please discuss this point

Line 168 - Can ultrasound be used for FNA of testicles in order to obtain sample for the microbiological examination?for drainage of abscesses? 

Line 179 - Torsion: this is an extratesticular disease that may produce testicular alteration. Please move it to the next paragraph (extratesticular diseases)

Line 209 - the term “very useful” is not specific. Please find citation in order to comment the usefulness of US in this topic (i.e. Felumlee et al, Vet radio Ultrasound 2012, vol 00, N 0, pp 1-5)

Line 212 - Please comment also the possibility of finding cryptorchid neoplastic testicles and provide their description. 

Line 256 - This sentence should refer to testicles. For other organs CT represent the imaging modality of choice rather than US (I.e liver)

Line 412 - Moreover CEUS can be performed on awake patients. No need for anesthesia. (In comparison to Ct or RMN)